# Autonomous Vehicle Dataset with Real Multi-Driver Scenes and Biometric Data

**DOI:** 10.3390/s23042009

**Published:** 2023-02-10

**Authors:** Francisca Rosique, Pedro J. Navarro, Leanne Miller, Eduardo Salas

**Affiliations:** División de Sistemas e Ingeniería Electrónica (DSIE), Campus Muralla del Mar, s/n, Universidad Politécnica de Cartagena, 30202 Cartagena, Spain

**Keywords:** autonomous vehicles, multimodal driving datasets, LiDAR, driver biometric data

## Abstract

The development of autonomous vehicles is becoming increasingly popular and gathering real-world data is considered a valuable task. Many datasets have been published recently in the autonomous vehicle sector, with synthetic datasets gaining particular interest due to availability and cost. For a real implementation and correct evaluation of vehicles at higher levels of autonomy, it is also necessary to consider human interaction, which is precisely something that lacks in existing datasets. In this article the UPCT dataset is presented, a public dataset containing high quality, multimodal data obtained using state-of-the-art sensors and equipment installed onboard the UPCT’s CICar autonomous vehicle. The dataset includes data from a variety of perception sensors including 3D LiDAR, cameras, IMU, GPS, encoders, as well as driver biometric data and driver behaviour questionnaires. In addition to the dataset, the software developed for data synchronisation and processing has been made available. The quality of the dataset was validated using an end-to-end neural network model with multiple inputs to obtain the speed and steering wheel angle and it obtained very promising results.

## 1. Introduction

Rapid advances in artificial intelligence, electronics, information and communications technology (leading to miniaturisation and improved performance of computers, sensors and networks) has led to the development of new approaches to Autonomous Vehicle technologies [1]. This together with new consumption habits and environmental awareness, where technology is vital and allows us to be more efficient and sustainable, has led to a considerable increase in the amount of research carried out on autonomous vehicles, making it the latest trend in the automotive industry [2]. Evidently, there is plenty of motivation and enthusiasm for speeding up progress, especially with the recent success of Big Data, Machine Learning and Deep Neural Networks.

Given the growing popularity of the development of autonomous vehicles, the collection of real data is considered a valuable task, with it being necessary for this sector to provide high-quality, multimodal and real-world datasets which can be used for benchmarking, simulation development, algorithms testing and diverse computer vision training exercises, among others.

The vehicle used for the data collection is usually equipped with a variety of sensors, such as cameras, Light Detection and Ranging (LiDAR) sensors, RADAR, GPS and Inertial Measurement Units (IMU). The raw data obtained by these sensors is recorded on a disk while the vehicle is being driven manually. Subsequently, the recorded data can be used to train and test different algorithms for autonomous driving, e.g., vehicle/pedestrian detection and tracking, Simultaneous Localization and Mapping (SLAM) and motion estimation [3].

In this context, many datasets have been published, a summary of the most popular datasets and their features is presented in Table 1. These datasets vary greatly in terms of traffic conditions, sensor configuration, application focus, data format, size, tool support, as well as other aspects. The most sought-after datasets dedicated to autonomous vehicle systems (AVS) are the so-called multimodal datasets. These datasets have gained particular attention recently, as datasets containing data from an individual sensor are insufficient to provide a complete perception of the environment. Furthermore, the most exploited sensors in this field, such as cameras, LIDARs, radars, etc., offer complementary data and their collaboration can guarantee a better understanding of the surroundings [3].

Table 1 provides a comparison of the main existing datasets, at both an academic and professional level and consists of a brief survey of datasets relevant to the development of autonomous driving systems. We focus on the most comparable and recent datasets, which strongly emphasise multimodal sensor data. Although they are not recent, we also include the KITTI and Udacity datasets as we consider them to be two of the most significant early driving datasets. We present the datasets in chronological order.

Despite the large number of existing studies, most of these datasets do not provide raw data, but instead offer labelled data to support training and evaluation, in particular semantic segmentation techniques. Obtaining real labelled data in large quantities is far from trivial. To start with, it is arduous and expensive to deploy multiple vehicles to collect images and data in a wide range of environmental, weather and lighting conditions. Secondly, the task of manually classifying each image is extremely time-consuming. Lastly, the accuracy of manually produced labels may be inconsistent across the dataset. These reasons, along with the level of fidelity achieved by 3D graphics engines, have encouraged the creation of synthetic datasets of artificial data based on scenes recreated by simulators [5].

As stated in the work by [19], this method of offering already labelled and even segmented data often presents problems in data quality due to the methods or models used. Another disadvantage of those models trained using only synthetic datasets is that in real-world scenarios, these tend to perform poorly, suffering from domain shift [20,21].

On the other hand, for a real implementation and correct evolution of autonomous vehicles at levels 4–5, it is also necessary to consider human interaction. Whether to infer a pedestrian’s intent to cross the road, identify a driver’s intent to perform a certain manoeuvre or detect potentially reckless moves, autonomous vehicles must have a high-level understanding of human behaviour. In most existing datasets, it is precisely this human data factor which is lacking. As can be seen in Table 1, apart from our proposal, the UPCT dataset, existing datasets dedicated to autonomous vehicles do not include biometric data or driver behaviour data.

In this article, we present the UPCT dataset, a public dataset of high-quality, multimodal data, obtained using state-of-the-art sensors equipped by the CICar autonomous vehicle belonging to the UPCT. The CICar includes sensors such as cameras, LiDAR, IMU, GPS and encoders, as well as biometric data from the drivers and driver behaviour questionnaires. The UPCT dataset offers the data acquired during 20 manual driving tests carried out by different drivers on an urban circuit, which consists of a circular route in the Spanish town of Fuente Alamo. To facilitate the use of the dataset, three large subgroups of data have been differentiated: Perception, Positioning and Driver data (biometrics and Driver Behaviour Questionnaire) and both the pre-processed raw data and the code which facilitates its use have been made available for download.

## 2. Materials and Methods

### 2.1. Experimental Design

To obtain the data, we decided to carry out ad hoc driving tests with a group of 50 healthy subjects of different ages and sex from the Region of Murcia (Spain), following the distribution shown in Table 2. The subjects were in possession of a valid type B driving licence (for driving cars, vans and, in general, vehicles with a maximum authorised mass of 3500 kg) at the time of the test. After performing the tests, the results of some subjects were excluded due to technical problems during the performance of the test or during the recording of the results, leaving a total of n = 20 subjects (11 male/9 female) with valid raw data to make up the final dataset.

#### 2.1.1. Driver Test Design

Before starting the experiment, in addition to the informed consent, each subject filled in two questionnaires: (1) the Biographic Questionnaire and (2) the Driver Behaviour Questionnaire.

The Biographic Questionnaire identifies key facts about the subject, such as gender, age and driving record.The Driver Behaviour Questionnaire (DBQ) collects self-reported data from the drivers, as there are no objective records of driving behaviour and previous traffic violations. The original DBQ consists of 50 items and is used to score the following three underlying factors: errors, violations, and lapses.

For this experiment, we have chosen to use the Spanish Driver Behaviour Questionnaire (SDBQ) [22], a shorter version adapted to Spanish drivers consisting of 28 items adapted to the peculiarities of the Spanish population. The version used consists of four factors, composed as follows: 6 traffic law violation items, 6 violation/aggressive manifestation items, 8 error items, and 8 lapse items. Participants were asked to indicate, on a 5-point scale, how often they had been involved in the behaviours or situations mentioned in the questionnaire.

#### 2.1.2. Driving Test Design

The driving test consists of one of the participating drivers, who has been equipped with a non-invasive smart band device, manually driving the UPCT-CICar vehicle (the equipment onboard and its characteristics will be explained in more detail in the following platform setup subsection) and following a previously established and identical route which is the same for all tests. Each driver had to complete one lap of the circuit, which included a parking exercise situated approximately halfway along the circuit.

The selected route is an urban circuit in the town of Fuente Álamo in the Region of Murcia, Spain, with the tests performed by multiple drivers manually driving the UPCT-CICar in real traffic situations (see Figure 1). This route provides a significant Point of Interest (POI) of typical urban driving situations: (a) intersections with priority and with “Give way”; (b) joining a roundabout, internal circulation and leaving the roundabout; (c) circulation in streets with “green wave” traffic lights; (d) traffic jams; (e) rapid incorporation to a high-density road through a side lane; and (f) pedestrian traffic on public roads.

In order to contemplate a variety of environmental and driving conditions, the tests were carried out at different times of the day (morning, afternoon or night). Figure 2 shows some images from the dataset, where different situations captured during the tests are shown.

After each driving test, the data acquired from the vehicle’s perception systems (LiDARs and cameras), positioning systems (IMU, GPS, rotation angle, acceleration, etc.) and biometric data from the driver are transferred to the central server.

#### 2.1.3. Platform Setup

For this work, the UPCT autonomous vehicle (UPCT-CICar [23]), was driven by a human pilot in manual mode. CICar is a real-world prototype, based on a commercial electric vehicle, the Renault Twizy, which has undergone a series of modifications to provide it with the required functionality. The CICar has been equipped with multiple sensors, including LiDAR, cameras, IMU, GPS, encoders, etc., necessary for the vehicle to perform autonomous driving tasks. This platform setup integrates a perception system, a control system, and a processing system on board the vehicle.

**Perception System**: The purpose of a sensor system is to collect data from the surrounding environment of the AV and send that data to the control system. These sensors measure different physical quantities, which are typically selected to overlap each other, providing the redundant information needed to correctly merge and correlate the information. In our autonomous vehicle, two types of sensors are used to measure the environment: short-range sensors (up to 10 m) and long-range sensors. Installed short-range sensors include a Sick 2D laser ranging scanner and time-of-flight camera. The long-range sensors are a 3D LIDAR scanner and a camera in the visible spectrum. Table 3 and Figure 3 show the different devices involved in data acquisition during the tests, as well as the details of the variables involved in obtaining them.

**Driver Biometric System:** The drivers’ biometric signal collection system has been carried out using a non-invasive wearable device, bracelet type, called Empatica E4. The Empatica E4 is a wrist-worn top-quality sensor device considered a Class IIa Medical Device according to 93/42/EEC Directive. Empatica E4 device measures the acceleration data (ACC), as well as other physiological parameters, namely the Blood Volume Pulse (BVP), from which the Heart Rate Variability (HRV) and the Inter-Beat Interval (IBI) are derived as well, skin temperature (TEMP) and also changes in certain electrical properties of the skin such as the Electrodermal Activity (EDA). For the creation of our dataset, among the several measurements recorded by the Empatica E4, this signal was considered, since it provides information better suited for activity recognition. A summary of the technical specifications of the accelerometer sensor is detailed in Table 4.

**Control System:** The main control systems of the Renault Twizy have been automated in order to allow the vehicle to be autonomously controlled. The modified systems are the steering wheel, the brake pedal and the accelerator pedal (see mechanical modification in Figure 3). Despite the fact that all driving will be manual and not autonomous, the system will record the data with two controller drives through a CAN bus. The Compact Rio cRIO 9082 controls the accelerator, brake and steering wheel movements with the CAN-Open communication protocol, as well as I/O signals.

**Processing System:** Each sensor works with its own sample rate, and in most cases, this is different between devices. The achieve the synchronisation of the data and accurately reconstruct the temporal sequence, time stamps have been generated to synchronise the operating start and finish times. All of this is controlled and synchronised by the on-board processing system.

## 3. Results

As a result of the different executions of the experiment with the participating subjects, a raw data set has been obtained that has been curated and published in a repository. The data in the repository is organized under three major directories: (1) Driver, (2) Perception and (3) Position. The distribution of the data in the different directories is detailed below.

### 3.1. Driver Directory

This directory contains information regarding the drivers, from the questionnaires completed before the test and the biometric data obtained during the test. The directory contains 20 Biometric_XX.csv files (one per driver, where XX is the driver identifier number) and a DBQ.csv file with the data collected from the Biographic Questionnaire and the Driver Behaviour Questionnaire forms.

For the composition of the Biometric_XX.csv files, a normalised sampling frequency of 4 Hz has been used and in the case of sensors with lower frequencies, the table has been completed with NaN fields, with the HR column being the only one affected as the sample rate of this field is 1 Hz. The Biometric_XX.csv files have the following table format, where each column contains the following information:(TIME): The first column corresponds to the time stamp expressed as a unix timestamp in UTC.(TEMP): Data from the temperature sensor expressed as degrees in Celsius (°C).(EDA) Measurement of electrodermal activity by capturing electrical conductance (inverse of resistance) across the skin. The data provided is raw data obtained directly from the sensor expressed in micro siemens (μS).(BVP) The BVP is the blood-volume pulse and the raw output of the PPG sensor. The PPG/BVP is the input signal to algorithms that calculate Inter beat Interval Times (IBI) and Heart Rate (HR) as outputs.(HR): This file contains the average heart rate values calculated at 10-s intervals. They are not derived from real-time readings but are processed after the data is loaded into a session.(ACC_X, ACC_Y, ACC_Z) Data from the three-axis accelerometer sensor. The accelerometer is configured to measure acceleration in the range [−2 g, 2 g]. Therefore, the unit in this file is 1/64 g. Data from x, y and z axis are displayed in the sixth, seventh and eighth columns, respectively.

The DBQ.csv file is made up of a total of 45 columns, where the first column contains the subject identifier. The rest of the columns correspond to each of the items from the Biographic Questionnaire and the Driver Behaviour Questionnaire forms, where the last 25 columns are the questions from the DBQ form.

### 3.2. Perception Directory

This directory contains:Twenty .bin type files, called perceptionXX.bin, where XX corresponds to the identifier number assigned to each driver at the time of the test.Twenty images from the RGB-D camera (front view).

In the .bin file, the data from the 3D LiDAR sensor, 2D LiDAR sensors and TOF cameras, obtained by the CICar perception system is saved. The data from these sensors were recorded continuously during the driving test, with data packets being written by the different sensors one after the other and at the exact moment in which they arrived at the system, without contemplating an established order of sensor reading and recording.

Each data packet consists of a header made up of two 32-bit integers, which identify the source of the data followed by the data as it was received from the sensors. The header format is as follows:

*uint32_t head [2]*;

head [0]: indicates the size in bytes of the packet received by the sensor.head [1]: contains a sensor identifier which shows the source of the data packet received.

The sensor identifiers are the following:


*//Packet identifiers in the data file*



*static const uint32_t LIDAR_PACKET_ID = 0 × 4C494452;  // 3D LiDAR*



*static const uint32_t GPS_PACKET_ID = 0 × 475053;   // GPS*



*static const uint32_t NMEA_STRING_ID = 0 × 4E4D4541;//*



*static const uint32_t M420_FRAME_ID = 0 × 4D343230;  // Camera ToF*



*static const uint32_t T551_FRONT_ID = 0 × 54354652;   // Front 2D LiDAR*



*static const uint32_t T551_BACK_ID = 0 × 5435424B;   // Rear 2D LiDAR*


The following is a .bin file example:


*(uint32_t) 1206 // Data packet size 1206 bytes*



*(uint32_t) 0 × 4C494452 // Data source: 3D LiDAR*



*(char [1206]) { ... } // 1206-byte vector with 3D LiDAR data*



*(uint32_t) 230,524 // Size of data packet 230,524 bytes*



*(uint32_t) 0 × 4D343230  // Data source: ToF camera*



*(char [230524]) { .. }   // 230524-byte vector with ToF data.*



*(uint32_t) 1206 // Size of data packet 1206 bytes*



*(uint32_t) 0 × 4C494452 //Data source: 3D LiDAR*



*(char [1206]) { .... }   //1206-byte vector with 3D LiDAR data.*



*(uint32_t) 1206 //Size of data packet 1206 bytes*



*(uint32_t) 0 × 4C494452  //Data source: 3D LiDAR*



*(char [1206]) { .... }   //1206-byte vector with 3D LiDAR data.*



*(uint32_t) 921  //Size of data packet 921 bytes*



*(uint32_t) 0 × 54354652  //Data source: Front 2D LiDAR*



*(char [921]) { ..... }   //921-byte vector with Front 2D LiDAR data.*


To facilitate the use and processing of the data, a programme has been developed that allows the data from each sensor to be extracted separately and independently into an additional .bin file. In this case, by separating the data into different files, the data packet identifier is not necessary, but synchronisation with the system is lost. Therefore, to avoid loss of synchronisation between the data packet of each sensor, the time stamp of the exact moment of capture must be included. The .bin file format for each independent sensor is as follows:


*uint32_t segundos    // Capture timestamp seconds*



*uint32_t microseg    // Capture timestamp microseconds*



*uint32_t numbytes    // Number of bytes in the data packet*



*char   datos[numbytes] // ‘raw’ data packet from the sensor*


This structure is repeated continuously for each data packet until the end of the file.

Furthermore, the data has been pre-processed and the 3D LiDAR, 2D LIDAR and ToF camera data from each test carried out They have been merged into a single point cloud and extracted to a .csv file called POINTCLOUD_XX.csv, where XX is the identifier assigned to each driver at the start of the test.

### 3.3. Position Directory

This directory contains information regarding the position system, obtained during the driving tests. The directory contains 20 Position_XX.csv files, one for each driver where XX is the driver identifier number. Each of these systems collects information from the GPS, IMU and Encoder sensors.

The Position_XX.csv files are saved in the following table format, where each column contains the following information:(TIME) This first column contains the timestamp of the session expressed as a unix UTC timestamp.(LATITUDE) latitude values obtained by the GPS.(LONGITUD) longitude values obtained by the GPS.(ALTITUDE) altitude values obtained by the GPS.(STERING_ANGLE): Steering wheel angle.(SPEED): Speed/(m/s).(DISTANCE_TRAVELLED).(LIN_ACEL_X): acceleration obtained around the *x*-axis, obtained in g.(LIN_ACEL_Y): acceleration obtained around the *y*-axis, obtained in g.(LIN_ACEL_Z): acceleration obtained around the *z*-axis, obtained in g.(ANG_VEL_X): angular velocity obtained around the *x*-axis, in degrees/second.(ANG_VEL_Y): angular velocity obtained around the *y*-axis, in degrees/second.(ANG_VEL_Z): angular velocity obtained around the *x*-axis, in degrees/second.

The acceleration or angular velocity values are given by four bytes. These bytes correspond to a real number according to the IEEE-754 standard. The IEEE-754 standard is the most widely used for the representation of floating-point numbers.

## 4. Technical Validation

### 4.1. Driver Test Validation

To validate the data regarding the drivers, the following actions were carried out:A first validation is carried out by measuring the reliability of the data obtained from the DBQS tests carried out on the drivers. The reliability of the questionnaires was obtained with the entire sample, finding Cronbach’s alpha indices and the two Guttman halves. The values to interpret the reliability were: <0.50 unacceptable; 0.50 ≥ poor < 0.60; 0.60 ≥ questionable/doubtful < 0.70; 0.70 ≥ acceptable < 0.80; 0.90 ≥ good < 0.90; and ≥0.90 excellent. The Cronbach Alpha coefficient is 0.796, which shows that the DBQ data set in this experiment has credibility [24].Missing data E4 data of seven participants (driver 1, driver 5, driver 17, driver 21, driver 30, driver 42, driver 45) were excluded due to a device malfunction during data collection. While physiological signals in the dataset are mostly error-free with most of the files complete above 95%, a portion of data is missing due to issues inherent to devices or a human error.

Raw data from the Empatica device was downloaded in csv format and analysed with the Kubios tool [25]. Kubios offers five artefact correction options based on very low to very high thresholds. No correction of the artefacts analysed by Kubios was necessary This is not surprising since the Empatica E4 already uses an algorithm that removes wrong IBIs or other wrong signals [26].

### 4.2. Driving Test Validation

Once the driving tests have been completed, a manual verification phase has been carried out on the data obtained (see Figure 4), where the data from those tests where reading or writing failures occurred, or failures in the test itself (routes, drivers, etc.) has been discarded.

Checking for abnormalities during the test. The time elapsed for the completion of each test has been checked, passing a filter, and discarding those tests in which the time has been either very short or too long. Data of five participants (driver 4, driver 17, driver 23, driver 29, driver 40) were excluded.Checking for errors in reading the sensors or writing to the disk. For each of the tests, the correct sending of information by the sensors during the test is verified. Those tests where a total or partial failure has been detected have been discarded. To detect these failures, the following aspects were checked:
a.All files exist on the disk. At the end of each test, the number of files generated has been checked. The absence of any of the files implies a failure to read or write the data occurred, therefore this test was discarded completely. Data of four participants (driver 1, driver 10, driver 31, driver 34) were excluded.b.Empty files. It has been verified that the files generated all contain data, discarding those tests where empty files have been detected. Data of two participants (driver 35, driver 36) were excluded.c.Exploratory data analysis. Considering the different types of data processed, different types of descriptive analytics have been chosen: (1) Analysis of data deviation. A standard deviation analysis has been applied to those data with discrete values (average speed, time travelled, etc.), discarding those data with a sharp deviation. Data of two participants (driver 11, driver 38) were excluded (2) Time series analysis: most of the data correspond to time series of data, with a certain variation of speed, for this reason, it has been decided to use the Dynamic Time Warping (DTW) technique.Checking for driving route failures. For each of the tests carried out, the route taken by the driver during the test has been verified, to make sure the driver stuck to the route initially stipulated. The test where a small deviation from the track occurred has been discarded. To verify this, the following checks were made: (1) steering wheel rotation pattern during the test, given that for the same trajectory the steering wheel rotation pattern must be similar for all the tests. (2) GPS trajectory, the trajectory has been painted and the tests that do not comply with the marked route have been eliminated.

After this first screening process, a quality validation of the resulting data is performed to guarantee the quality of the data (see Figure 4). Our validation method comprised three steps: (1) Quality control of variables. (2) Quality control of support media. (3) Experimental validation.

#### 4.2.1. Quality Control of Variables

An analysis of the internal structure of the set of circumstances of the DBQ form (content validity) has been performed. To be able to apply a factorial analysis correctly, those items with a declaration frequency of less than 5% were eliminated. Subsequently, and since the items on the form are dichotomous variables, the tetrachoric correlation coefficient was applied to obtain the correlation matrix between the 28 items.

The reliability of the questionnaires was obtained with the entire sample, finding Cronbach’s alpha indices and the two Guttman halves. The values to interpret the reliability were: <0.50 unacceptable; 0.50 ≥ poor < 0.60; 0.60 ≥ questionable/doubtful < 0.70; 0.70 ≥ acceptable < 0.80; 0.90 ≥ good < 0.90; and ≥0.90 excellent. The Cronbach Alpha coefficient is 0.796.

Secondly, outliers in the acquired data, those values notably different compared to the patterns present in the rest of the data, may be due to errors in reading and writing the data from the sensors. Certain deviations were detected in the data from the GPS, due to momentary loss of signal or where the position has been calculated with a fewer number of satellites. In those cases in which the losses are less than two consecutive time intervals, a prediction of the vehicle’s position is made. For cases where the loss is greater, the tests have been discarded. To apply this prediction a constant acceleration Kalman filter has been used.

#### 4.2.2. Quality Control of Support Media

The clocks of all the sensors and devices were synchronised at the start of each experimental test session. All devices are controlled by the control unit on board the vehicle, which provides a perfect temporal and spatial synchronisation of the data obtained by the sensors (see Figure 5).

It has been verified that the data obtained by the encoder is synchronised with the rest of the sensor data. This was achieved by checking the distance indicated by the encoder coincides with the distance calculated between two consecutive GPS timestamps (GNSS). This was done using the Havershine expression shown in Equation (1), where d is the distance in metres between two points on the Earth’s surface; r is the Earth’s radius (6378 km); *φ*_1_ and *φ*_2_ are the latitudes in radians; Ψ_1_ and Ψ_2_ are the longitudes in radians of two consecutive timestamps.
(1)$$d=2r\sin^{-1}\left(\sqrt{\sin^{2}\left(\frac{\varphi_{2}-\varphi_{1}}{2}\right)+\cos\varphi_{1}\cos\varphi_{2}\sin^{2}\left(\frac{\Psi_{2}-\Psi_{1}}{2}\right)}\right)$$

The indirect data provided by the encoder has also been verified, for example, that the speed matches the direct data measurements provided by the GPS.

#### 4.2.3. Experimental Validation

Finally, the most conclusive validation was performed: the usability analysis of the data contained in the final dataset. The work by Navarro et al. [20] presents the implementation of six end-to-end deep learning models trained using the UPCT dataset. The different end-to-end models were tested using different data sources from the vehicle, including RGB images, linear accelerations and angular velocities. We trained two models using only RGB image data, two using both the image data and IMU data as input to the models, and the last two used sequences of images as an input.

The best results were obtained using a mixed data input type end-to-end deep neural network model which used the front images obtained by the vehicle camera and angular speeds from the IMU to predict the speed and steering wheel angle, obtaining a mean error of 1.06%. An exhaustive optimization process of the convolutional blocks has demonstrated that it is possible to design lightweight end-to-end architectures with a high performance more suitable for the final implementation in autonomous driving.

## 5. Conclusions

In this work, we have presented the UPCT dataset, a real-world public driving dataset with 20 sets of driving data from 20 drivers which performed a driving test on an urban circuit in real traffic situations. The dataset contains different types of data which we have divided into three categories: (1) Driver, (2) Perception and (3) Position.

The dataset has been validated and tested with six end-to-end deep neural network models, using the RGB image data and IMU data, obtaining very promising results considering the size of the dataset. The detailed results are published in the work by Navarro et al. [20]. We plan to continue this research by making use of the depth images and comparing the results to those obtained when using just RGB images, as well as performing data augmentation to increase the sample sizes.

The main novelty of this dataset is the collection of biometric driver data which allows the behaviour of autonomous driving models to be compared to human drivers. In future research, we plan to use biometric driver data to perform driver behaviour studies. An interesting approach would be to relate the stress levels of the driver to certain driving situations, such as entering a roundabout, entering a main road or parking, for example. As each of the 20 drivers completed the same circuit, it would also be possible to compare the different driving styles and relate these tendencies to certain age groups or to a particular sex. In addition, in the driving behaviour questionnaire, each driver was asked about their driving style, and with the driving test, this can be compared to their real-life performance to determine if drivers correctly perceive their attitudes whilst driving.

## Figures and Tables

**Figure 1 sensors-23-02009-f001:**
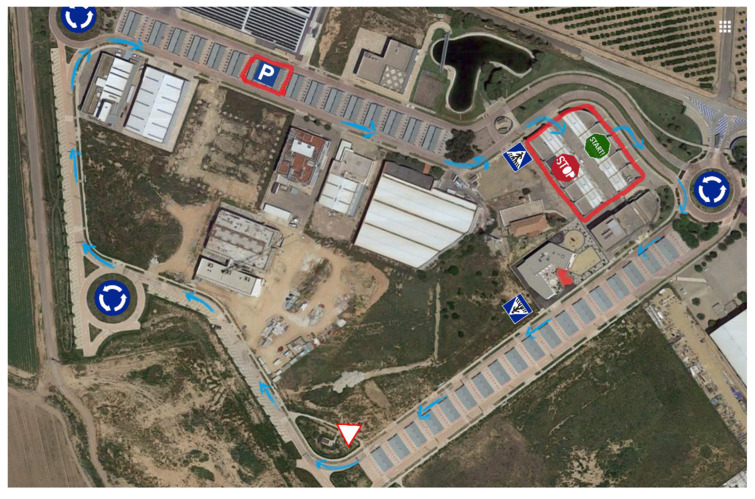
Urban route selected for the driving tests.

**Figure 2 sensors-23-02009-f002:**
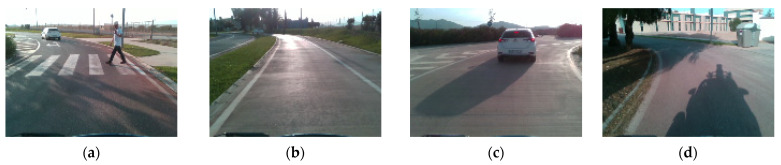
Images from dataset. (**a**) Pedestrian crossing; (**b**) saturation due to reflections on the road; (**c**) car braking; (**d**) complex shadows on the road.

**Figure 3 sensors-23-02009-f003:**
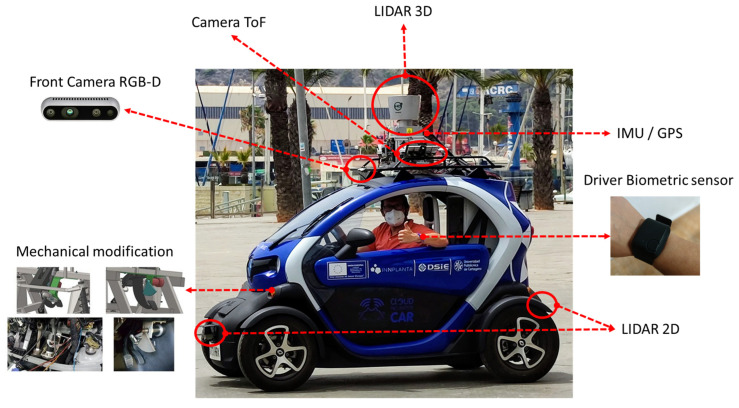
Autonomous vehicle UPCT-CICar and its different sensors and devices.

**Figure 4 sensors-23-02009-f004:**
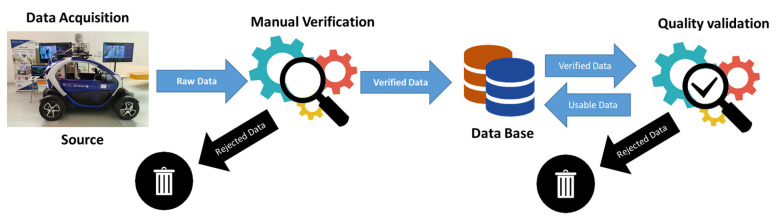
Verification process.

**Figure 5 sensors-23-02009-f005:**
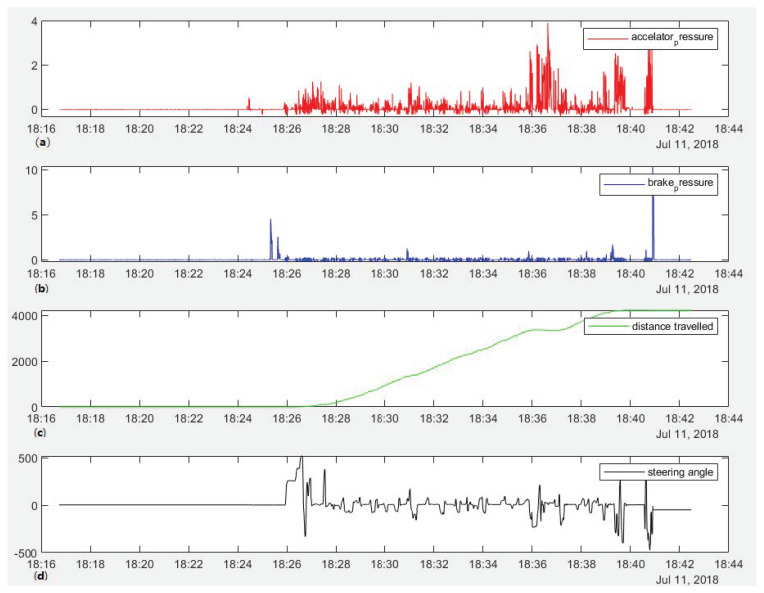
Data synchronisation with timestamp. (**a**) accelerator; (**b**) break; (**c**) distance; (**d**) steering wheel angle.

**Table 1 sensors-23-02009-t001:** Comparison of the main Datasets.

Ref./Year	Samples	Image Type	LIDAR	RADAR	IMU/GPS	Control Actions	Raw Data	Driver Data	Real Data	Biometrics Data	Driver Behaviour
UPCT	78 K	RGB, Depth	Yes	No	Yes	Steering wheel, Speed	Yes	Yes	Yes	Yes	Yes
KITTI [4]/2012	15 K	RGB	Yes	No	Yes	-	Yes	No	Yes	No	No
Udacity [5]/2016	34 K	RGB	Yes	No	Yes	Steering wheel	Yes	No	No	No	No
Lyft L5 [6]/2019	323 K	RGB	Yes	No	Yes	-	Yes	No	Yes	No	No
nuScenes [7]/2019	1.4 M	RGB	Yes	Yes	Yes	-	Partial	No	Yes	No	No
Pandaset [8]/2019	48 K	RGB	Yes	No	Yes	-	Partial	No	Yes	No	No
Waymo [9]/2019	1 M	RGB	Yes	No	Yes	-	Yes	No	Yes	No	No
PreSIL [10]/2019	50 K	RGB	Yes	No	No	-	No	No	No	No	No
GAC [11]/2019	3.24 M	RGB	No	No	No	Steering wheel, Speed	N/A	No	Yes	No	No
A2D2 [12]/2020	392 K	RGB	Yes	No	Yes	Steering angle, brake, accelerator	Partial	No	Yes	No	No
IDDA [13]/2020	1 M	RGB, Depth	No	No	No	-	No	No	No	No	No
Appollo Scape [14]/2020	100 K	RGB	Yes	No	No	-	No	No	Yes	No	No
Cityscapes [15]/2020	25 K	RGB	No	No	Yes	-	No	No	Yes	No	No
OLIMP [16]/2020	47 K	RGB	No	Yes	No	-	Yes	No	Yes	No	No
PixSet [17]/2021	29 K	RGB	Yes	Yes	Yes	-	No	No	Yes	No	No
ONCE [18]/2021	1 M	RGB	Yes	No	No	-	No	No	Yes	No	No

**Table 2 sensors-23-02009-t002:** Demographic distribution of subjects by gender and age.

	Categories	n Initial	% Initial	n Final	% Final
**Gender**	Male	26	52	11	55
	Female	24	48	9	45
**Age**	18–24	6	12	3	15
	25–44	22	44	10	50
	45–64	17	34	5	25
	>=65	5	10	2	10

**Table 3 sensors-23-02009-t003:** Sensor data in CICar.

Device	Variable	Details
LiDAR 3D	Scene	Long-range sensors 3D High-Definition LIDAR (HDL64SE supplied by Velodyne) Its 64 laser beams spin at 800 rpm and can detect objects up to 120 m away with an accuracy of 2 cm 1.3 Million Points per Second Vertical FOV: 26.9°
2 × LiDAR 2D	Scene	Short-range sensors Sick laser 2D TIM551 Operating range 0.05 m–10 m Horizontal FOV 270° Frequency 15 Hz Angular resolution 1° Range 10% of reflectance 8 m
2 × ToF	Scene	Short-range sensors ToF Sentis3D-M420Kit cam Range: Indoor: 7 m, Outdoor: 4 m Horizontal FOV: 90°
RGB-D	Scene	Short-range sensors Depth Camera D435 Intel RealSense range 3 m Up to 90 fps Depth FOV: 87° × 58° RGB FOV: 69° × 42°
IMU	Localisation, longitudinal and transversal Acceleration	NAV440CA-202 Inertial Measurement Unit (IMU) 3-axis accelerometer Bandwidth: 25 Hz Pitch and roll accuracy of <0.4°, Position Accuracy < 0.3 m
GPS	Localisation	EMLID RTK GNSS Receiver 7 mm positioning precision
Encoder	Distance	
Biometric sensors	Driver Biometric signals	Empatica E4 EDA Sensor (GSR Sensor), PPG Sensor, Infrared Thermopile 3-axis Accelerometer

**Table 4 sensors-23-02009-t004:** Biometric variables details.

Variable	Sampling Frequency	Signal Range [Min, Max]	Details
ACC	32 Hz	[−2 g, 2 g]	Accelerometer 3 axes data (x, y, z).
EDA	4 Hz	[0.01 µS, 100 µS]	Electrodermal activity by capturing electrical conductance (inverse of resistance) across the skin.
BVP	64 Hz	n/a	Blood Volume Pulse.
IBI	64 Hz	n/a	Inter-beat interval (obtained from the BVP signal)
HR	1 Hz	n/a	Average Heart Rate (obtained from the BVP signal). Values are calculated at 10-s intervals.
TEMP	4 Hz	[−40 °C, 115 °C]	Skin Temperature.

## Data Availability

The data set presented here, which is available within Figshare and released under a CC-BY 4.0 license. https://figshare.com/s/4b9a25a958c3ec578362 (accessed on 1 December 2022).

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
