# Peer review of "Autonomous Vehicle Dataset with Real Multi-Driver Scenes and Biometric Data"

_sensors, 2023, doi:10.3390/s23042009_

Round 1
Reviewer 1 Report
This paper present a new dataset related to multi-driver scenes and 2
biometric data.
- Keywords are not completed.
- Related works are poor and not up to dated.
- There is no conclusion at all.
Author Response
We appreciate your comments and recommendations, in the following pdf file we respond to your suggestions and observations.

Reviewer 2 Report
The authors of the paper presented research in the field of public collection of high-quality, multimodal data, obtained using state-of-the-art sensors with which an autonomous vehicle is equipped, and most importantly biometrics data and driver questionnaires.
In their study, the authors extracted a perception system whose purpose for collect data from the AV environment and transfer this data to the control system. The driver's biometric system, which is responsible for collecting drivers' biometric signals base was implemented using a non-invasive device. All the data is analyzed by processing system.
In the paper, the authors clearly presented the methodologies of the implemented research and the results of the obtained research. They also conducted a verification of the proposed solution.
The article is understandable and worth publishing in a journal. However, I recommend that the authors should add a short summary - conclusions and plans for further research. The reader, after reading this good article, is interested in conclusions of the manuscript.
Author Response

(The authors gave the same response as above.)

Reviewer 3 Report
The authors present a public dataset containing multiple sensors and driver biometric data and driver behavior questionnaires. It is a valuable task. But this paper focuses too much on the description and generation of the dataset. So it lacks a little academic content. My suggestions are below.
1. Please show more details on the potential usage of this dataset. For example, training which type of neural network or using it in which scenario can achieve a more advantageous result.
2. The characteristics of the selected route for collecting the data should be discussed. Many readers may think the complexity of the route is not enough.
3. The section about the file type of the dataset is too long.
4. Only one experiment in reference [18] cannot support the importance of the dataset. In my opinion, some concise applications and achievements should be added to show the dataset’s value.
Author Response

(The authors gave the same response as above.)

Round 2
Reviewer 1 Report
This version is quite better than previous. I think it is ready for publication.
Reviewer 3 Report
The authors have revised the paper and answered my queries. Thanks.